# Self-assembling nanofibrous bacteriophage microgels as sprayable antimicrobials targeting multidrug-resistant bacteria

Lei Tian [1], Leon He [1], Kyle Jackson [1], Ahmed Saif[1], Shadman Khan [2], Zeqi Wan[1], Tohid F. Didar [2,3,4] & Zeinab Hosseinidoust [1,2,3] ✉

Nanofilamentous bacteriophages (bacterial viruses) are biofunctional, self-propagating, and monodisperse natural building blocks for virus-built materials. Minifying phage-built materials to microscale offers the promise of expanding the range function for these biomaterials to sprays and colloidal bioassays/biosensors. Here, we crosslink half a million self-organized phages as the sole structural component to construct each soft microgel. Through an in-house developed, biologics-friendly, high-throughput template method, over 35,000 phage-built microgels are produced from every square centimetre of a peelable microporous film template, constituting a 13-billion phage community. The phage-exclusive microgels exhibit a self-organized, highly-aligned nanofibrous texture and tunable auto-fluorescence. Further preservation of antimicrobial activity was achieved by making hybrid protein-phage microgels. When loaded with potent virulent phages, these microgels effectively reduce heavy loads of multidrug-resistant *Escherichia coli* O157:H7 on food products, leading to up to 6 logs reduction in 9 hours and rendering food contaminant free.

Bacteriophages, also known as phages, are natural bacterial predators and their job in nature is to keep bacterial populations in check[1,2]. Phages infect bacteria in a highly targeted manner—some are able to identify and kill a single strain of bacteria in a heterogeneous population. It follows that when used for biocontrol in environments with a pre-existing commensal bacterial populations, such as certain food products or applications in agriculture, farming, or human therapeutic use, phages are less likely to disturb the delicate balance of such communities while still being able to eliminate harmful bacteria[3]. Foodborne diseases result in hundreds of thousands of deaths each year, almost a third of whom are young children[4]. Phage products have been approved by the US Food and Drug Administration for control of dangerous bacterial contaminants such as *Escherichia coli*, *Salmonella*, or *Listeria* in food products[5,6]. The use of phage for food safety has the distinct advantage that, unlike most antimicrobials, it will not impact the taste, texture, and nutritional quality of the food, and can be safely applied to decontaminate food products from farm to market to consumer plates[7,8]. However, widespread use is still limited. This is at least partly due to challenges in delivery and stability, which in turn limit the efficacy of the phage products[9].

Nature makes phages in a variety of shapes and sizes[10]. Phages are, in essence, proteinous nanoparticles that encase a genome, enabling the propagation of wild-type or genetically modified virions into a suspension of identical and monodisperse nanoparticles[11]. In addition, the phage surface chemistry can be customized with atomic precision via genetic engineering or chemical conjugation, making phage virions a powerful building block for multifunctional antimicrobial material[12–14].

We have previously reported a simple chemistry that was effective at crosslinking filamentous phage, yielding bulk soft material displaying the basic properties of a hydrogel[15–17].

[1]Department of Chemical Engineering, McMaster University, Hamilton, ON L8S 4L7, Canada. [2]School of Biomedical Engineering, McMaster University, Hamilton, ON L8S 4K1, Canada. [3]Michael DeGroote Institute for Infectious Disease Research, McMaster University, Hamilton, ON L8S 4K1, Canada. [4]Department of Mechanical Engineering, McMaster University, Hamilton, ON L8S 4L7, Canada. ✉e-mail: doust@mcmaster.ca

Herein, we reinvented phage gels by developing a high-throughput manufacturing method that not only enabled generation of microgels made up entirely of viral nanoparticles, but also preserved the bioactivity of the phage building blocks in the process. Compared to polymeric microgels such as poly(N-isopropylacrylamide)[18] and poly(ethylene glycol)[19], phage microgels remain unexplored, partly because of challenges in manufacturing such microgels. Common microgel preparation methods such as microfluidics[20] or the emulsion method[21] are not suitable for microgels encapsulated with or made from heat/solvent-sensitive chemicals or biomolecules (such as proteins and viruses) that must retain their bioactivity through the preparation process.

Microgels offer major distinct advantages over bulk material. Namely, they have larger surface areas and thus more contact points for phage with contaminating bacteria as well as enhanced flow properties in suspensions, allowing for delivery via spray or injection, all of which make them a more versatile option for biocontrol in environmental, food, and medical applications[22–24]. Packing phages into soft, hydrated material further has the advantage of preservation against desiccation and harsh environments[25,26]. The hydrated structure of microgels offers the advantage of preserving desiccation-sensitive biomolecules.

We report a biomolecule-friendly, high-throughput method to synthesize detachable phage microgel arrays, where a microporous mold, made through the breath figure method, was used without the need for large equipment or complex infrastructure. We demonstrate that two crosslinkers can each effectively assist the gelation of phage nanofilaments through different crosslinking reactions, leading to vastly different fluorescence profiles. We further show that the phage nanofilaments in these virus-exclusive microgels self-assemble into an orderly, highly aligned nanofibrous structure that serve as a high-load vehicle for protein and strong virulent phages to control multidrug-resistant *E. coli* O157:H7 in food products.

## Results and discussion
### Generation of bacteriophage microgels

The gelation of phage aqueous suspension is based on the crosslinking reaction between M13 filamentous phage and a small molecule chemical crosslinker. The crosslinker, glutaraldehyde (GA), can react with multiple functional groups on the phage coat protein, notably amino groups on the lysine residues[15,27,28]. As shown in Fig. 1a, reaction 1, a GA molecule reacts with 2 amino groups on two phage capsids and forms Schiff bases connecting these two phages. It is worth mentioning that GA in aqueous solution is not limited to regular monomeric formation. For example, cyclic hemiacetal and cyclic hemiacetal oligomer are common forms of GA which can react with amine as well and form ether groups (Supplementary Fig. 1). All these possible reactions proceed simultaneously leading to a crosslinked network of phage nanofilaments. The formation of these side products, in addition to the well-documented self-polymerization of GA and the strong autofluorescence of the final gel[15], make GA a non-ideal choice for certain applications. Therefore, we explored a second small molecule crosslinker 1-Ethyl-3-(3-dimethylaminopropyl) carbodiimide (EDC), which proved capable of crosslinking phage through a different crosslinking mechanism[29]. An EDC molecule first reacts with a carboxyl group on the phage coat protein and forms an amine-reactive intermediate that quickly reacts with an amine group on another phage coat protein to form an amide bond between two phages (Fig. 1a, reaction 2)[30,31]. It is noteworthy that EDC molecules were not incorporated into the final crosslinked product. Instead, EDC took oxygen atoms from carboxyl groups on phages and formed water-soluble isourea by-product which can be easily washed out. This intermediate role of EDC is the fundament of fabricating phage-exclusive microgels. This mechanism was found to be effective at controlling the optical properties of the microgels, as will be discussed later, which potentially brings different applications to phage microgels.

A single M13 phage exhibits abundant amine and carboxyl groups (8100 and 10,800, respectively) on its protein coat (Supplementary Note 1)[32], providing rich reaction sites for crosslinking reactions. The crosslinked phage virions form a network, resulting in the gelation of the phage aqueous suspension. We observed that M13 and EDC mixture needs less than 12 h to gel while the same concentration of GA takes about 24 h to gel.

It is noteworthy that heat and organic solvents are commonly involved during microparticle preparation or isolation[23,33], which have irreversible detriments for biomaterials. Therefore, manufacturing viral microgels without losing bioactivity would have been exceptionally challenging without developing a suitable microgel manufacturing method. Herein, we proposed a biomaterial-friendly approach for the parathion of pure and hybrid phage microgel. The phage microgels were gelled in and isolated from a polystyrene honeycomb film containing uniform open-ended spherical micropores throughout the film surface, as illustrated in Fig. 1b.

The honeycomb films here were prepared via a well-established approach known as the breath figure method[34,35]. This is an easy-approachable and rapid method to fabricate the large-scale template with single-layer, closely-packed, and homogenous micropores without any large equipment. The size of the template can be changed by applying different volumes of polystyrene solution to the glass slide. In the current experiments, 600 μL of polystyrene solution can generate a honeycomb film with a diameter of 2.5 cm in 20 min. The micropores were uniformly rounded and the cross-sectional Scanning Electron Microscope (SEM) image in Fig. 1c indicates that the inner pores exhibited open-ended spherical shape. The inner diameter of the template pores was $35.73 \pm 2.86$ μm, as measured by electron micrographs.

To fabricate phage microgels, a mixture of M13 suspension and crosslinker (GA/EDC) was cast on a plasma-treated polystyrene honeycomb film where the mixture fills inside the micropores (Fig. 1b). After removing the redundant phage solution on the template surface by a glass slide, the film was transferred into a sealed humid container at 4 °C for 1 day for gelation. As shown in Fig. 1d, the original phage suspension inside the pores successfully turned into an ordered array of solid phage microgels. The honeycomb template is a thin round polystyrene film, which means the composite film loading the phage microgel array is flexible. Figure 1e shows a composite film (phage microgels inside the template film) tailored into a 1 cm² square which can easily bend. These properties make the film an excellent patch integrating phage microgels for further antimicrobial and biosensing studies.

Moreover, the microgel array inside the template is detachable. A piece of adhesive tape was used to stick on the composite honeycomb film surface to then peel off the top half of the pores (Supplementary Fig. 2a). Consequently, the top half of the pores was attached on the tape and the microgels inside the film were exposed on the bottom film layer without damage (Fig. 1f). Phage microgels were conveniently isolated from the template by immersing the film in sterile water and sonicating. Figure 1g shows the permeable polystyrene pore network on the tape and the shallow pore structure left on the honeycomb film after peeling and sonication. The microgels were detached successfully and transferred to water phase. Supplementary Figure 2b reveals the edge of the pealing area where the top left of the image is the peeled area of the honeycomb film showing shallow pores and the bottom right area is the unpeeled deep pores. The peeling procedure effectively separated the honeycomb film (Supplementary Fig. 2c, d). Phage microgels were suspended in Millipore water. Figure 1h, i shows the isolated phage aerogel microbeads which

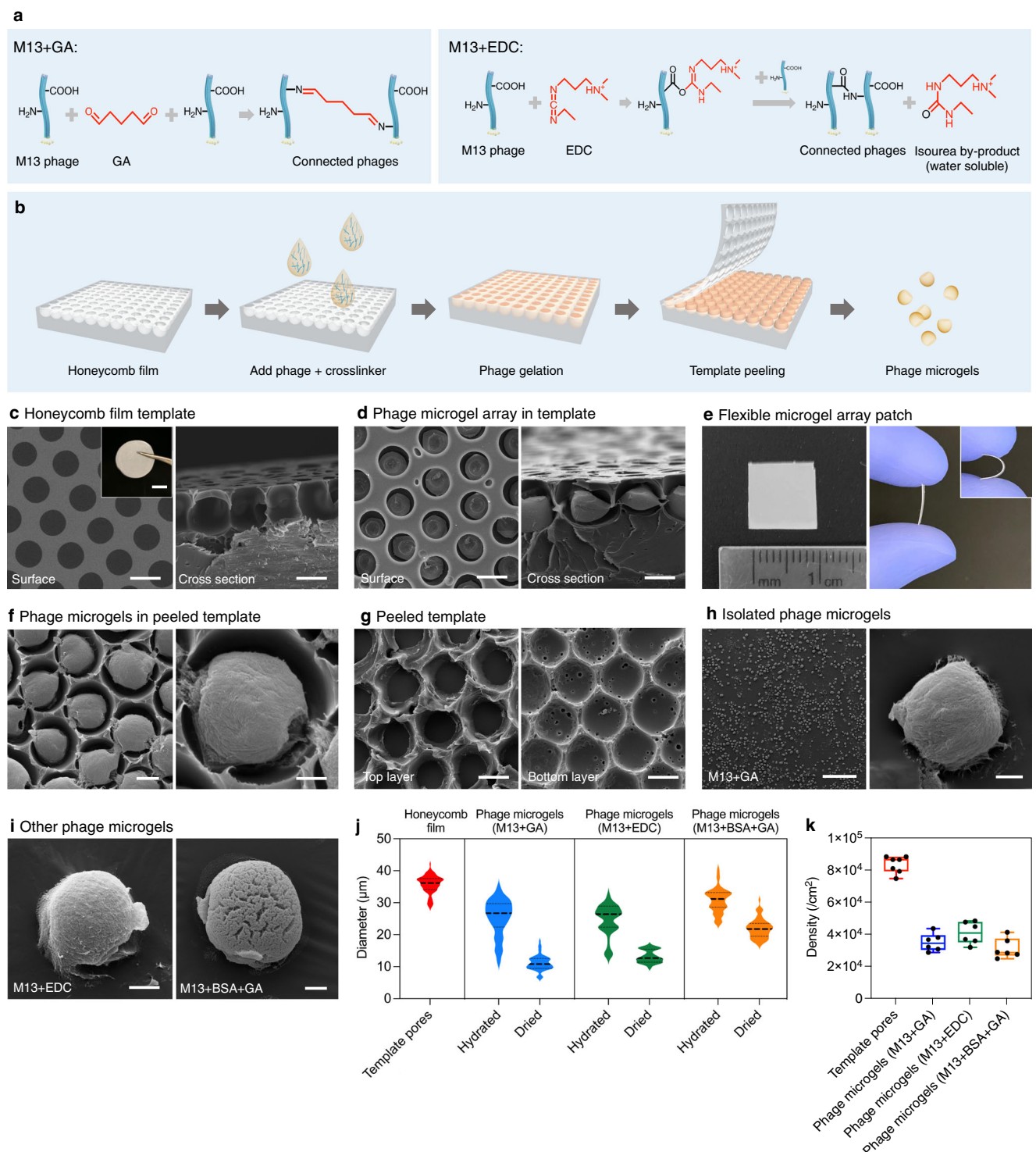

are the M13 + GA/EDC microgels after critical point drying, proving the microgel array inside the template are detachable. In addition to pure phage microgels, we also added bovine serum albumin (BSA) to M13 + GA solution to demonstrate application of the developed method to fabricate phage-protein hybrid phage microgels, which can further expand the functionality. BSA efficiently provided abundant amino groups and carboxyl groups to consume excessive crosslinker molecules and preserve the bioactivity of M13 phages, which will be illustrated in Supplementary Fig. 7. In addition, the gelation reaction is accelerated from over 12 h to about 30 min. Although template

methods have been used for making microparticles in the past with calcination[35], we essentially reinvented the manufacturing method to enable production of microscale colloidal soft matter, namely phage microgels, in the form of a peelable patch or a suspension. This method is high-throughput, heat-free, and solvent-free, which makes it especially advantageous to keep biomolecules active. The size and shape of the microparticles is determined to the template pore shape and size. Fortunately, there are already abundant studies extending the breath figure method to fabricate honeycomb films containing ordered pores at different sizes and shapes[36–38].

**Fig. 1 | Preparation of bacteriophage microgels using honeycomb film as template. a** Crosslinking reactions of M13 bacteriophages with glutaraldehyde (GA) and 1-Ethyl-3-(3-dimethylaminopropyl) carbodiimide (EDC), respectively. **b** Schematic image of preparing phage microgels: Honeycomb film is plasma-coated to increase hydrophilicity; Phage and crosslinker mixture solution is cast on the film and placed in the vacuum for 10 min; The film is placed in a humid environment in 4 °C for 2 days; The top layer of the film is peeled off using adhesive tape; The phage microgels are isolated after removing the film. **c** Surface and cross-section image of the polystyrene honeycomb film surface (scale bar 20 μm). Insert: photo of the round polystyrene honeycomb film (scale bar 1 cm). **d** Surface and cross-section SEM images of M13 crosslinked by GA inside the pores of honeycomb film ($n = 3$ independent experiments. Scale bar 20 μm). **e** A flexible honeycomb film containing phage microgel array. Photos of an independent composite film tailored into a 1 cm² square, further showing the film can easily be bent by hand. **f** SEM images of phage microgel array left on a peeled honeycomb film (scale bar 10 μm) and zoom-in SEM image of a single phage microgel in this array (Scale bar 5 μm). **g** SEM images of the permeable pore network on adhesive tape after peeling (left) and the honeycomb film after peeling and sonication (right). Scale bar 20 μm. **h** SEM images of the isolated phage microgels (scale bar 500 μm) and a single phage microgel made with M13 phages and GA (scale bar 5 μm). **i** SEM images of a phage microgel made with M13 crosslinked with EDC (left) and GA + BSA (right). Scale bar 10 μm. BSA: bovine serum albumin. $n = 3$ independent experiments for **d**–**i**. **j** Size distribution of the template pores ($n = 84$ pores measured over 3 independent templates) and the phage microgels prepared with GA ($n = 58$ and 57 microgels at hydrated and dried status respectively, measured over 3 independent experiments), EDC ($n = 45$ and 53 microgels at hydrated and dried status respectively, measured over 3 independent experiments) and BSA ($n = 56$ and 54 microgels at hydrated and dried status respectively, measured over 3 independent experiments). (Violin plot lines indicate 25th, 50th, 75th percentile). **k** Pore density of the template ($n = 7$ independent films) and the produced microgel count from every square centimeter of the template ($n = 6$ independent experiments for each type of microgels). Box plots show minimum to maximum (whiskers), 25–75% (box), median (band inside) with all data points.

## Size distribution, porosity and preparation efficiency of phage microgels

Figure 1j presents the size distribution of the template pores and phage microgels. The M13 phage microgels crosslinked by GA and EDC separately show a similar size distribution, which is smaller than the template pores ($25.34 \pm 5.72$ and $24.39 \pm 4.92$ μm, respectively). It is noteworthy that the size range of the microgels has a broader distribution compared to the template pores. This might be caused by shrinkage during gelation and possible breakage in the isolation process. Based on the phage concentration used ($5 \times 10^{13}$ PFU mL$^{-1}$) and the average microgel size, we can estimate that each phage microgel is composed of over $7 \times 10^5$ M13 phages. The average size of phage microgels containing BSA is larger ($30.77 \pm 3.83$ μm) and the size range is narrower. The addition of BSA likely makes the microgels denser so the shrinkage and damage during the whole process is reduced, which is reflected in the average size and size distribution.

The porosity of these phage microgels was evaluated by measuring the size change between hydrated and air-dried states. The GA and EDC microgels decreased in diameter to $11.13 \pm 2.32$ μm and $13.16 \pm 1.986$ μm after dehydration, showing 91.5% and 84.3% volume reduction, respectively. The high-volume reduction of phage microgels suggests high porosity. The microgels with added BSA had significantly less shrinkage, maintaining an average size of $21.97 \pm 3.04$ μm (63.60% volume size reduction), indicating denser, less porous microgels.

The preparation efficiency of the phage microgels was investigated by calculating the microgel count obtained from every square centimeter of the template (details in Supplementary Note 2, Supplementary Fig. 3). The honeycomb film template contained $83,862 \pm 5241$ micropores cm$^{-2}$. The usage of GA crosslinker produced $35295 \pm 5490$ phage micropores cm$^{-2}$ while EDC crosslinker produced $41,226 \pm 6878$ micropores cm$^{-2}$ (Fig. 1k). The addition of BSA produced similar results ($31,431 \pm 6185$ micropores cm$^{-2}$). The number of microgels produced from the template is lower than the template pore density (42.1%, 49.2%, and 37.5%). This could be a result of partial filling of phage suspension into the pores or the loss during isolation process. In summary, we obtained over $3.5 \times 10^4$ phage microgels from every square centimeter of our template where each microgel contains more than $3.8 \times 10^5$ phage particles, constituting a phage community of $10^{10}$ in total. Every film we made was over 5 cm², allowing for the production of 175,000 phage microgels in a single day. In addition, more than 10 films can be applied to produce microgels simultaneously, demonstrating the high-throughput ability of this method.

## Highly aligned nanofibrous texture of phage microgels

As shown in Fig. 2a, M13 phages are high aspect ratio nanofilaments (length = 880 nm, width = 6.6 nm) with a relatively large tip, where five copies of the bacterium-binding protein (g3p) protrude from one end. The size of the small molecule crosslinker GA ($M_w = 100.11$ g mol$^{-1}$) and EDC ($M_w = 191.70$ g mol$^{-1}$) is negligible in comparison. We hypothesized that the phage microgels have nanofibrous texture which are crosslinked filamentous phages, as shown in the schematic image in Fig. 2b.

As shown in the SEM images (Fig. 2c), the M13 aerogel microparticles crosslinked by GA showed a sophisticated thread pattern. At higher magnification, we observed nanofibers aligning at a single orientation overall where these nanofibers (width between 7 nm and 20 nm) fit the width of a single M13 phage. The uneven width of fibers could likely be a result of the inevitable 3 nm Pt coating for SEM or the occasional lateral binding of multiple phages. The M13 aerogel microparticles crosslinked by EDC showed a similar ordered nanostructure (Fig. 2d). The bright dots in these images are the g3p bacterium-binding sites; these are large protruding proteins that stand out in electron micrographs. In summary, FE-SEM imaging showed densely packed, self-assembled nanofilaments that formed highly-ordered nanofibrous network during the gelation process.

It is noteworthy that the phage alignment in the hybrid phage-protein (M13 + BSA) microgels crosslinked by GA was distinct. The phage nanofilaments in the hybrid microgels were partially embedded in BSA with no particular order (Fig. 2e). Adding BSA to the microgel system accelerated the gelation process from 12 h to 30 min, leaving insufficient time for phages to self-assemble. The g3p proteins are still exposed on the microgel surface, providing bacterium-binding sites. Moreover, we investigated pure BSA protein microgels crosslinked by GA to compare the nanostructure with that of phage microgels and confirmed that the observed order in the M13 microgels was caused by the M13 nanofilaments and not by the crosslinker or the dehydration procedures. The BSA microgels were processed by the same preparation and dehydration procedures and showed no sign fibrous nanostructures. The SEM images of these microgels (Supplementary Fig. 4) shows irregularly rough surfaces, unlike the nanofibrous texture of the phage microgels.

In conclusion, the phage-exclusive microgels exhibit high porosity, potentiating their strong loading capacity of proteins, phages, and small molecules. The homogenous nanofibrous texture along the same orientation is the direct evidence that the microgels are composed by phages solely crosslinked by small molecule. The addition of protein interfered the order alignment of phages, but played an important role in preserving the phage bioactivity which will be illustrated later.

## Autofluorescence of phage microgels can be tuned by using different crosslinkers

As shown in Fig. 3a, the phage microgels made with GA showed significant autofluorescence in four channels. This phenomenon is

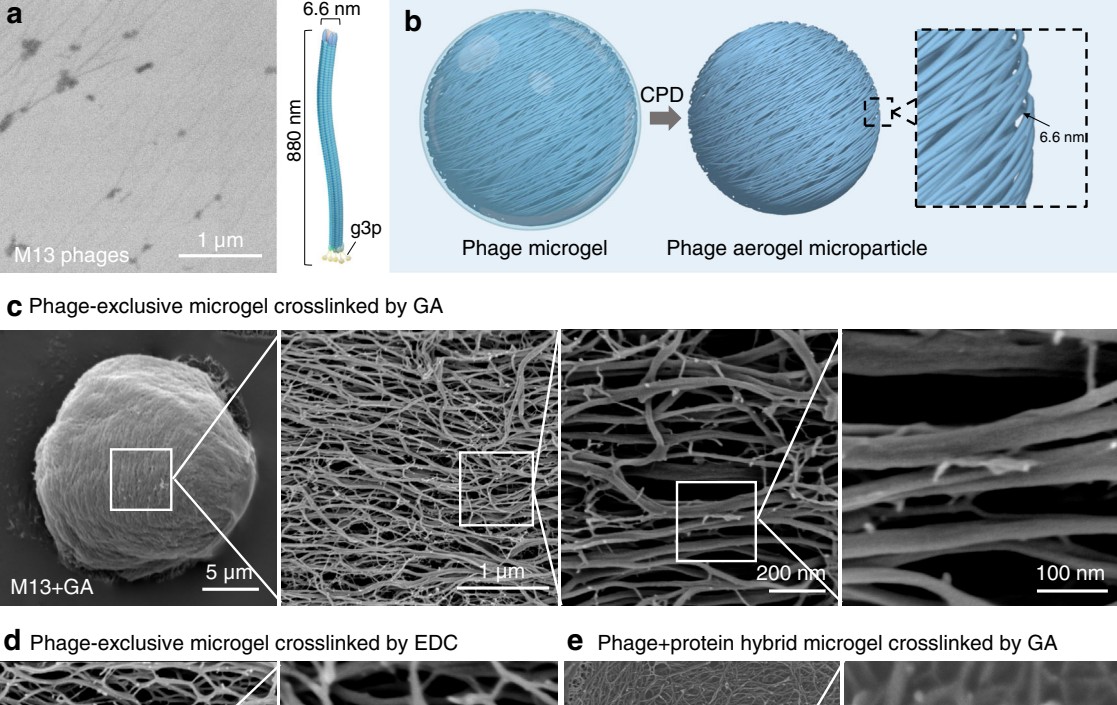

**Fig. 2 | Aligned nanofibrous texture of phage microgels. a** SEM image of non-crosslinked M13 phage nanofibers and schematic image of a nanofilamentous M13 phage showing high aspect ratio. **b** Schematic image of a hydrated phage microgel composed of crosslinked filamentous M13 phages turning into a phage aerogel microparticle by critical point drying (CPD), showing a nanofibrous texture. **c** SEM images of a M13 phage microgel crosslinked by GA and the highly aligned nanofibrous texture on the microgel surface. **d** SEM images of the orderly aligned nanofibrous texture on a M13 phage microgel crosslinked by EDC; **e** SEM images of the nanofibrous texture on a M13 + BSA microgel cross-linked by GA. (*n* = 5 independent experiments for **a** and **c**–**e**. GA glutaraldehyde, EDC 1-Ethyl-3-(3-dimethylaminopropyl) carbodiimide, BSA bovine serum albumin).

associated with electronic transitions such as the *n*-π* transitions of C = N in the Schiff's base generated from crosslinking reactions[15,39]. This autofluorescence potentiates non-destructive imaging capability of the microgels. However, fluorescence of phage microgels can be troublesome in some application scenarios, for example certain bio-sensing applications that rely on fluorescence to detect the target analytes. For this reason, we explored microgels of phage crosslinked with an alternative small molecule crosslinker, namely EDC, that do not exhibit autofluorescence (Fig. 3b). The amide bonds formed between M13 phage and EDC cause a much weaker fluorescent signal, thus expanding the range of applications for the microgels. In the green and orange channels, the fluorescence of M13 + GA and M13 + BSA + GA microgels was 294.7% and 320.9% higher than M13 + EDC microgels. The largest difference appeared in the red channel where GA-crosslinking microgels had fluorescent signal but the fluorescence of EDC-crosslinking microgels was not observed. Both GA- and EDC-crosslinking phage microgels showed low-fluorescence in the blue channel, and the strongest fluorescence was observed in the orange channel (94.7% and 82.5% higher than their second strongest channel, green).

For scenarios where a strong fluorescence signal is anticipated to be advantageous, we can add BSA to M13 + GA microgels to participate in gelation to enhance the fluorescent signal (Fig. 3c). The quantified fluorescent intensity of those microgels is illustrated in Fig. 3d. The

addition of BSA followed the same trend at different channels as pure phage microgels, and enhanced the fluorescent intensity compared to the GA microgels (23.5% higher at green channel and 26.1% higher at orange channel). Fourier transform infrared (FTIR) spectra further confirmed the functional groups on different phage microgels (Fig. 3e). The spectra of M13 + GA, M13 + EDC and M13 + BSA + GA microgels are very similar because of the abundant functional groups on the proteinous capsid of phages. All three spectra showed typical peaks at 1660, 1530, and 1230 cm⁻¹, representative for the amide I, II, and III bonds on protein capsids. The three unique peaks showing in GA-crosslinking microgels rather than EDC-crosslinking microgels are at around 3050, 2950, and 1450 cm⁻¹ corresponding to sp²C–H stretch, aldehyde C–H stretch, and imine (C = N) bonds respectively.

In addition, we monitored the gelation procedure using fluores-cence microscopy to confirm that the fluorescence signal is the result of gelation and not inherited from the phage building blocks, tem-plates, or crosslinkers. During the gelation process, a distinct change in fluorescence was observed using microscopy with four different optical filter sets. As shown in Fig. 3f–h, 5 × 10¹³ PFU mL⁻¹ of M13 phage suspension with 0.1 M of either crosslinker inside the honeycomb film showed no fluorescence. After the gelation of phage with GA was complete (with or without the addition of BSA), the composite films emitted obvious fluorescent signal at green, orange, and red channels. The signal in blue channel was very weak. This change is also observed

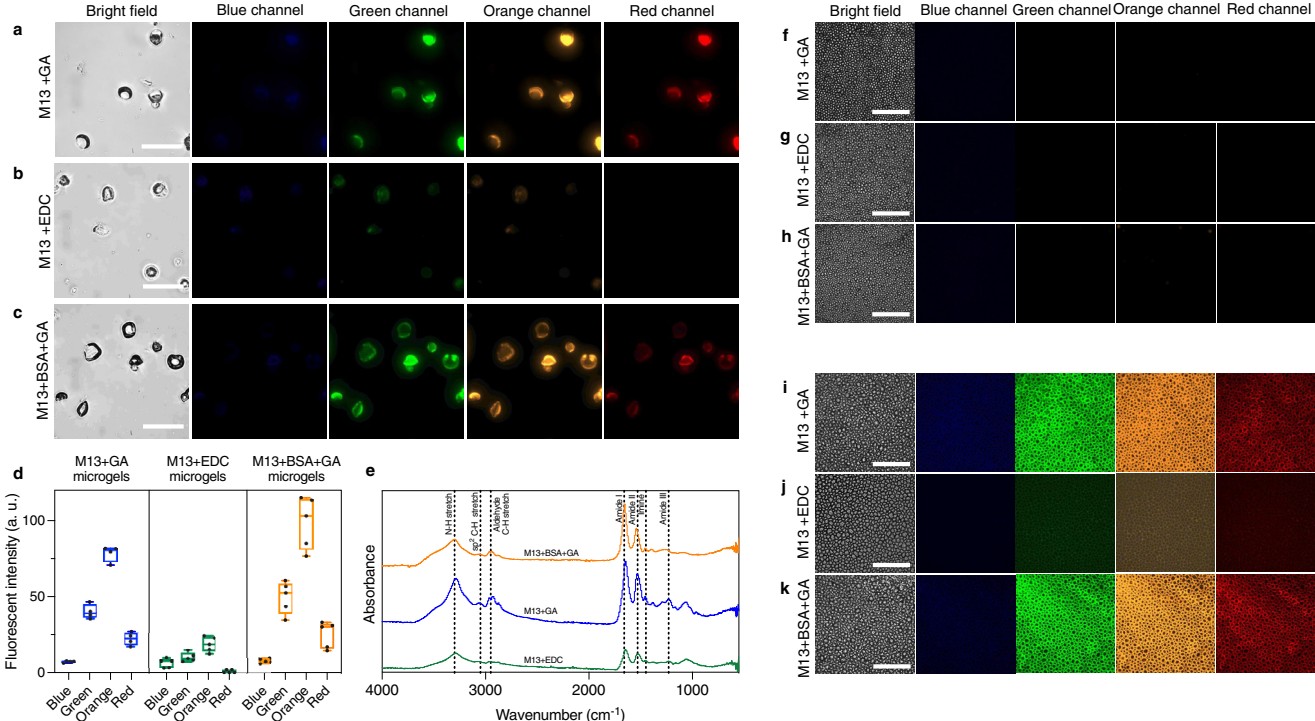

**Fig. 3 | Fluorescence profile of M13 phage microgels and fluorescence profile of the template before and after the microgel formation. a–c** Fluorescent images of three types of phage microgels made of $5 \times 10^{13}$ PFU mL$^{-1}$ of M13 phage with **a** 0.1 M GA, **b** 0.1 M EDC, and **c** 2% BSA + 0.1 M GA. Scale bar: 100 µm. GA glutaraldehyde, EDC 1-Ethyl-3-(3-dimethylaminopropyl) carbodiimide, BSA bovine serum albumin. **d** Quantified fluorescent intensity of 3 types phage microgels under four different channels. Box plots show minimum to maximum (whiskers), 25–75% (box), median (band inside) with all data points. a.u. arbitrary unit. M13 + GA microgels: $n = 4$ microgels per fluorescent channel. M13 + EDC and M13 + BSA + GA microgels: $n = 5$ microgels per fluorescent channel. **e** FTIR spectra of phage microgels. **f–h** Fluorescent images of honeycomb template filled with the mixture solution corresponding to **a–c**. Scale bar: 500 µm. **i–k** Fluorescent images of corresponding honeycomb films after the gelation of phage solution inside. Scale bar: 500 µm. Different fluorescent channels are: 1, bright field; 2, film excited at 340 nm and imaged using a $\lambda = 435$ nm optical filter (blue channel); 3, film excited at 465 nm and imaged using a $\lambda = 515$ nm optical filter (green channel); 4, film excited at 528 nm and imaged using a $\lambda = 590$ nm optical filter (orange channel) and 5, film excited at 625 nm and imaged using a $\lambda = 670$ nm optical filter (red channel). ($n = 3$ independent experiments for **f–k**).

in the composite film with EDC where its fluorescence signal at these four channels is significantly lower (Fig. 3i–k). This observation confirms that the crosslinking reactions are the sole reason for autofluorescence and suggests that using different crosslinkers can aid in fine-tuning the fluorescence signal to match the requirements of different application scenarios.

### Targeted antimicrobial functions of phage microgel patches and sprays

We hypothesized that the phage microgels inherited the antimicrobial activity of their phage building blocks and are able to specifically target host bacteria. To investigate the antimicrobial performance of pure and potentially hybrid phage microgels, we used M13 along with two virulent *E. coli* phages, namely T7 and HER262, which have strong and specific killing action but different geometric shapes and mechanisms of infection (details in Fig. 4a, Supplementary Note 3, Supplementary Fig. 5a). Further, we demonstrated that the hydrated environment in microgels can protect desiccation-sensitive phages. As shown in Supplementary Fig. 5b, the titers of phages M13, HER262 and T7 all decreased about 4 logs after drying for 1 h at room temperature and rehydrating.

We demonstrated antimicrobial activity of phage microgels in three biocontrol scenarios: an undetached microgel array in the template as an antimicrobial patch, a microgel sprayer (Supplementary Fig. 6a), and the addition of microgels directly to a bacterial-contaminated liquid. The microgel spray directly used our microgels suspension, containing over $3 \times 10^4$ microgels mL$^{-1}$ and all microgels

were washed twice. Free phage was not detectable in the eluent after this point. Initially, we evaluated the infectivity of pure M13 phage microgels. It was found out that these two types of microgels did not show obvious infectivity to *E. coli* ER2738 (M13 phage's natural host), neither as a patch or spray (Supplementary Fig. 6b). The lack of obvious bioactivity stems mainly from the fact that M13 has a low antimicrobial activity even in free suspension form[10]. Intramolecular crosslinking can further decrease this already low activity. On the contrary, M13 + BSA + GA hybrid microgels maintained their infectivity and the corresponding patch formed lysis zone around the edges on a lawn of *E. coli* ER2738 and sprayed microgels formed plaques on the bacteria lawn (Supplementary Fig. 6b), clearly indicating antimicrobial activity. The non-infectivity of BSA + GA microgels confirmed that the bioactivity of M13 + BSA + GA microgels is not bound to BSA or reacted crosslinker (Supplementary Fig. 6b). Therefore, it is possible that the abundant amino groups and carboxyl groups offered by BSA consumed excessive crosslinker molecules, minimizing the intramolecular crosslinking within phages and eventually protecting the bioactivity of phages. FE-SEM imaging showed the bacterial-binding sites, g3p, were displayed on the surface of phage microgels, so these microgels were expected to bind multiple *E. coli* ER2738 as shown in the schematic image Fig. 4a. Even though M13 + BSA microgels successfully retained the bioactivity, the hybrid phage microgels cannot kill *E. coli* sufficiently considering that M13 is a weakly antimicrobial phage to begin with[40]. Supplementary Fig. 6c shows that adding M13 + BSA microgels to a ER2738-contaminated nutrient environment cannot prevent the growth of *E. coli*, again indicating that a stronger virulent phage should

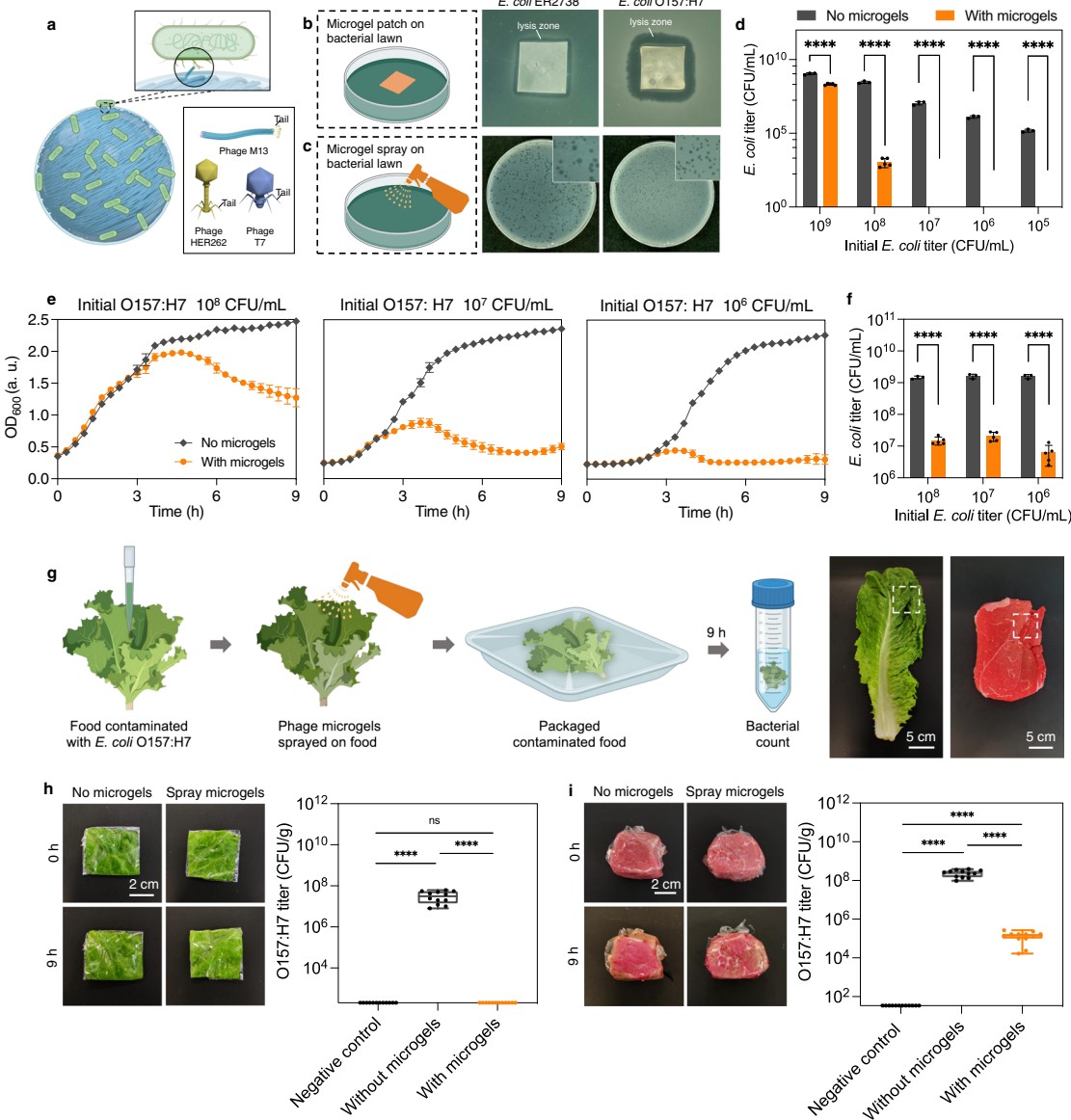

**Fig. 4 | Antimicrobial activity of hybrid phage microgels. a** Schematic image of a phage microgel where the M13 g3p is binding to the tip of the F pilus on the host *E. coli*. Box on the bottom right: Comparison of the shapes of phage M13 (filamentous), HER262 (long tailed), T7 (short tailed). **b** Photos of the HER262-embedded hybrid phage microgels patch arrays forming lysis zones on the lawn of both *E. coli* ER2738 and *E. coli* O157:H7. **c** HER262-embedded hybrid phage microgels sprayed on the lawn of both *E. coli* ER2738 and O157:H7, showing clearing zones on both lawns. **d** Titer count of *E. coli* O157:H7 incubated in PBS after 9 h at different initial concentration with (*n* = 5 independent experiments per bacterial concentration) and without (*n* = 3 independent experiments per bacterial concentration) HER262 microgels. (*p* < 0.0001 for all 5 bacterial concentrations). **e** Kill curves for *E. coli* O157:H7 suspension, incubated in TSB for 9 h at different initial concentration with and without HER262 microgels (*n* = 3 independent experiments per bacterial concentration). **f** Final titer count of the *E. coli* O157:H7 incubated in TSB with (*n* = 5 independent experiments per bacterial concentration) and without

(*n* = 3 independent experiments per bacterial concentration) HER262 microgels after 9 h in part **e**. **g** Left: Schematic image of microgel sprays decontaminating multidrug-resistant *E. coli* O157:H7 in lettuce. Right: pictures of lettuce and meat. White boxes indicate where we cut the lettuce and meat into small pieces. **h** Left: pictures of wrapped artificially contaminated lettuces sprayed with water and microgels, respectively, at 0 and 9 h. Right: bacterial titer count of the collected bacterial suspension from contaminated lettuces (*n* = 12 independent lettuces per group). **i** Left: pictures of wrapped artificially contaminated meats sprayed with water and microgels, respectively, at 0 and 9 h. Right: bacterial titer count of the collected bacterial solution from the contaminated meat (*n* = 12 independent meat per group). Date are presented as mean ± SD. Box plots show minimum to maximum (whiskers), 25–75% (box), median (band inside) with all data points. Statistical significance is derived from unpaired *t*-test in panel **d**–**f**, and two-way analysis of variance (ANOVA) in panel **h**–**i**. ****$p$ < 0.0001. Schematics created with BioRender.com.

be used for bactericidal activity. It is noteworthy that M13 is an attractive phage for use as structural component (mainly due to its shape and readily available toolkits for genetic engineering) but is not commonly the phage of choice for biocontrol of bacterial contamination/infections where bactericidal activity is desired.

Integrating phages as microgels was expected to offer four main advantages compared to applying a phage suspension for biocontrol. One is desiccation control (Supplementary Fig. 7). Another is that

microgels can achieve very high local concentration. We calculated that based on the titer of phage suspension needed to fabricate microgels ($5 \times 10^{13}$ PFU mL$^{-1}$), each microgel contains more than $3.8 \times 10^5$ of M13 phages. This not only benefits the antimicrobial application, but also provides massive recognition sites when using modified phages to construct microgel biosensors. Additionally, the nanofibrous structure of M13 phage microgels provides strong loading capacity to load other antimicrobial factors, such as antibodies, small

molecule inhibitors, or other bacteriophages. Finally, microgels offer a higher surface area compared to a bulk microgel, thus increasing the contact area between phage and its bacterial host which is expected to increase the antibacterial potency of the microgel over the same weight of bulk hydrogel.

## Hybrid phage microgels targeting multidrug-resistant bacteria

We embedded our microgels with strong virulent phages to enhance the bactericidal ability of our microgels. Virulent phages, a class of phage with strong antimicrobial action, are different in physical structure and mechanism of antibacterial action than filamentous phage. Preserving the antibacterial action of virulent phages inside the gels, however, is a major challenge because their host recognition/binding sites are often located asymmetrically on tip of their tail fibers (such as phage HER262 and T7 which were previously shown in Fig. 4a). The relatively fragile tail fibers can be easily damaged through processing, or blocking. We addressed this challenge by not only optimizing the chemistry, but also by working at the microscale, thus increasing the surface area for phage action.

To minimize the intramolecular crosslinking within phages, we decreased the concentration of GA from 0.1 M to 0.02 M. At this low concentration of crosslinker, the phage suspension cannot gel without the presence of BSA. The first virulent phage we added to the M13 + BSA microgels was phage HER262 ($1 \times 10^{10}$ PFU mL$^{-1}$) that targets multidrug-resistant *E. coli* O157:H7, a common bacterial contaminant on meats and lettuces[41,42]. The SEM images confirmed the formation of hybrid microgels (Supplementary Fig. 8). Comparing to the capsulated nanofibrous structure of M13 + BSA + GA microgels (Fig. 2e), the surface nanostructure of M13 + HER262 + BSA + GA microgels displayed nanodots matching the capsid size of phage HER262 (~50 nm), supporting retention of HER262 antimicrobial activity through its ability to target *E. coli* O157:H7. As shown in Fig. 4b, the unseparated hybrid phage HER262 microgels in the patch formed lysis zones on the lawns of both *E. coli* ER2738 and *E. coli* O157:H7. The microgel formed clear plaques, indicative of antimicrobial activity, when sprayed on both lawns (Fig. 4c).

To evaluate the bacteria-killing ability of our microgels in liquid, we incubated *E. coli* O157:H7 and phage microgels together in two environments: phosphate-buffered saline (PBS) simulating a nutrient-deficient environment, and nutrient tryptic soy broth (TSB) simulating a nutrient-rich environment.

For the nutrient-deficient environment, we used PBS to dilute the pre-culture into $10^8$, $10^7$, $10^6$, and $10^5$ CFU mL$^{-1}$. Phage microgels were then added to the diluted bacterial suspensions at a final concentration of ~1500 microgels mL$^{-1}$. As shown in Fig. 4d, *E. coli* O157:H7 maintained the same concentration level in PBS without microgels after 9 hrs. On the contrary, when microgels were added, they killed all bacteria within 9 h when the initial concentration of *E. coli* O157:H7 was below $10^7$ CFU mL$^{-1}$, and decreased the concentration 6 logs when initial concentration was high, $10^8$ CFU mL$^{-1}$.

In nutrient TSB, phage microgels also showed the ability to prevent bacterial growth, but at a much faster rate. This is expected because the phage antimicrobial activity is closely tied to the physiological state of the host bacteria. We used TSB to dilute the pre-culture into $10^8$, $10^7$, $10^6$ CFU mL$^{-1}$ and monitored the optical density at a wavelength of 600 nm (OD$_{600}$) in the suspension to evaluate bacterial growth. As shown in Fig. 4e, the phage microgels restrained the increase of OD$_{600}$ in 4 h regardless at high and low contamination loads. We further quantified the bacterial titers after 9 h (Fig. 4f). Phage microgels prevented the growth of *E. coli* O157:H7, maintaining the bacterial titer between $10^6$ and $10^7$ CFU mL$^{-1}$ while all the controls reached $10^9$ CFU mL$^{-1}$.

In summary, phage microgels displayed excellent antimicrobial ability regardless of nutrient in the environment, especially in the nutrient-deficient environment where bacterial propagation was inhibited. Moreover, to demonstrate specific bactericidal activity of our microgels, we incubated phage microgels with ER2738 and BL21 at the initial titer of $10^6$ CFU mL$^{-1}$ and the bacterial solutions showed same strong growing trend regardless of the participation of microgels (Supplementary Fig. 9), illustrating the specific targeting of these microgels.

We confirmed that the antimicrobial activity was independent of the phage used as long as a virulent phage was used, by demonstrating the results with the virulent phage T7 (Supplementary Fig. 10a). The patch and spray made with T7-embedded phage microgels showed lysis zones on the lawn of BL21 (Supplementary Fig. 10b). The microgels can decrease the titer of BL21 by at least 6 logs (Supplementary Fig. 10c). In nutrient TSB solution, T7-embedded phage microgels can still prevent the bacterial growth and caused a 5 log difference (Supplementary Fig. 10d, e), which proved the strong function of phage microgels as delivery vehicle.

## Phage microgel spray for food product safety

After verifying the antimicrobial activity of HER262-embedded phage microgels against *E. coli* O157:H7, we used these microgels to inhibit bacterial contamination in two completely different food matrices. As illustrated in Fig. 4g, the lettuce was first contaminated with *E. coli* O157:H7 at $10^6$ CFU g$^{-1}$, followed by spraying with phage microgels. The lettuce was then covered with food wrap and placed in room temperature for 9 h. The second day, the lettuce was immersed in 10 mL of PBS and vortexed for 2 min to collect the live bacteria. As shown in Fig. 4h, we cannot visually differentiate between the lettuce leaves with different treatment, but the bacterial concentration in the collected solution is significantly different. The contaminated lettuce with no microgel treatment reached an average contaminant load of $3.3 \times 10^7$ CFU g$^{-1}$ after 9 h. For the contaminated lettuce sprayed with microgels, the bacterial titer dropped to undetectable level (<100 CFU g$^{-1}$, up to 6 log reduction). The same antimicrobial phenomenon was observed when testing artificially contaminated meat samples with a similar treatment. The meat samples showed no visual difference. The O157:H7-contaminated meat sprayed with water reached $2.5 \times 10^8$ CFU g$^{-1}$, For the contaminated meat sprayed with microgels, the bacterial titer dropped to $1.4 \times 10^5$ CFU g$^{-1}$, indicating that the microgels killed 99.94% of the drug-resistant bacteria.

In conclusion, the main discoveries and contributions of our work are (1) establishment of virus-built microparticles, (2) development of a biomolecular-friendly high-throughput preparation method for diverse phage microgels, (3) highly aligned self-organized nanofibrous texture of phage-exclusive microgels, (4) tunable autofluorescence, and (5) the application of the phage-protein hybrid microgel patch and microgel sprays for biocontrol. The high-throughput method we propose here combined with honeycomb template casting with peel isolation. It produced over 35,000 phage microgels in every square centimeter template with each microgel containing half a million phages. This method can be extended to prepare most types of microgels efficiently, but it is particularly suited to heat/solvent-sensitive microgels as it is simple, heat-free, and solvent-free, which is especially useful to keep biomolecules and proteinaceous materials functional. Our nanofilamentous building blocks self-assembled forming a highly aligned nanofibrous structure where single phage filaments could be observed using an electron microscope. Addition of BSA protein in microgels added additional flexibility in design, namely, to tune the fluorescence and preserve phage bioactivity. Furthermore, strong virulent phages were combined into the microgels and the resulting microgel patch, microgel spray and microgel suspension were proven highly effective in their antimicrobial action. Specifically, the contaminant load of the multidrug-resistant *Escherichia coli* O157:H7 in food products were reduced by 6 logs after spraying phage microgels. We further demonstrated that aside from packing a high density of antimicrobial virions, the microgels also protected against

desiccation. Every year, it is estimated that 600 million people fall ill due to the consumption of contaminated food. This attributes to 420,000 annual deaths globally and *E. coli* contamination is considered a major factor[43]. Incorporating our antimicrobial microgels or patches into packaging, sprays in grocery store produce sections, and in household decontaminating products can effectively inhibit bacterial contamination in a human-friendly manner that will ultimately reduce foodborne illnesses, deaths, and associated economic loss.

## Methods

### Preparing polystyrene honeycomb film
Honeycomb films were prepared using the breath figure method[35,44]. This high-throughput method does not require any large equipment or any premade microparticles as templates. Briefly, 600 μL of 5 wt% of polystyrene (Mw = 650 000, Millipore Sigma) in chloroform was cast and spread circularly on a clean glass slide in a humid chamber (~55% relative humidity monitored by a humidity sensor). The chamber was sealed immediately after adding the polystyrene solution to maintain humidity. After 20 mins, the polystyrene solution had solidified, forming a white film on the glass slide. The slides were then taken out of the chamber. After 1 h, the honeycomb film was easily peeled off and stored at room temperature.

### Phage propagation, purification, and concentrating
M13 bacteriophage was propagated using its host: *Escherichia coli* strain K12 ER2738 (New England Biolabs Ltd., E4104S). A pre-culture of *E. coli* was prepared in LB-Miller broth and placed in a shaking incubator overnight set to 180 rpm and 37 °C. The following day, 2.5 mL of the pre-culture was added to 250 mL of LB broth in a baffled flask. Subsequently, a 10 μL aliquot of M13 phage ($10^{12}$ PFU mL$^{-1}$) was added to the flask to initiate the propagation. The flask was incubated in a shaking incubator set to 180 rpm and 37 °C for 5 h. Fifty milliliters aliquots of propagated phage solution were then centrifuged at $7000 \times g$ for 15 min. The resulting bacteria pellets were discarded, and the phage-containing supernatant was stored at 4 °C.

The purification of the propagated M13 phage supernatant was achieved through an aqueous two-phase polyethylene glycol (PEG) precipitation protocol followed by an ultracentrifugal filtration process[45]. A 20 (w v$^{-1}$) % PEG solution was aseptically prepared and supplemented with 2.5 M NaCl solution. The sterile PEG solution was added in a 1:6 ratio to the propagated phage supernatant and incubated in a fridge overnight at 4 °C. Subsequently, the incubated PEG-phage solutions were centrifuged at 4 °C and $5000 \times g$ for 45 min to obtain pelleted phage. The resulting phage was then resuspended in 5 mL of sterilized water and incubated overnight on a roller at 4 °C. The resuspended phage was then centrifuged at $5000 \times g$ for 15 min to remove the remaining bacterial contaminants. The described PEG/NaCl purification procedure was subsequently repeated a second time to ensure all contaminants were removed. The resulting PEG-purified phage solution was then filtered through centrifugal filters (MWCO 100 KDa and 30 KDa, Millipore Sigma, Ultra-15) to remove excess water. The final concentration was titered using plaque assay method[46].

### Phage microgel preparation
A polystyrene honeycomb film was used as the template to prepare the phage microgels. The film was initially plasma-coated with $O_2$ for 5 min and then covered with 100 μL of the mixture of M13: $5 \times 10^{13}$ PFU mL$^{-1}$ with GA or EDC: 0.1 M. The film was subsequently placed inside a desiccator connected to a vacuum pump. The pump was turned on for 5 min to create a low-pressure environment which helped the phage solution fill inside the micropores. The film was then taken out and transferred into a sealed humid container at 4 °C for 1 day.

After 2 days, a glass slide was used to remove the excess phage hydrogel on the template surface. After this cleaning step, a piece of transparent adhesive tape was adhered to the template film surface and then peeled off to remove the top layer of the template. Then the template film was immersed in 1 mL of sterilized water or PBS and sonicated for 10 min. After the sonication, the film was taken out and discarded. The microgels were suspended in water and stored at 4 °C for further experiments.

### Scanning electron microscopy
Samples were pre-treated using the critical point drying method to dehydrate the microgels without disturbing their surface nanostructures. Samples were processed through an ethanol gradient treatment and then placed in a Leica critical point dryer (EM CPD300) for 3.5 h.

Two types of Scanning Electron Microscopy (SEM) were used to image the templates and microgels. TESCAN VEGA-II LSU SEM was used to image these samples, where 10 nm layers of gold were coated onto the samples in advance. A field emission scanning electron microscope (FEI Magellan 400) was used to image the nanostructure on the surface of the microgels, where 3 nm layers of Pt were coated onto the samples in advance.

### Inverted fluorescence microscopy
An inverted microscope (Nikon Eclipse Ti2 inverted microscope) was used to take bright field and fluorescent images of the microgels and their templates. Four different optical filter sets (blue channel: ex/em = 340/435 nm; green channel: ex/em = 465/515 nm; orange channel: ex/em = 528/590 nm; red channel: ex/em = 625/670 nm) were used for fluorescence imaging. The excitation filter was positioned in front of the LED light source, and the image was captured using the emission filter attached to the camera. The intensity of the light source and the exposure time were consistent.

### Size measurement of template pores and microgels
An inverted microscope (Nikon Eclipse Ti2) was used to image the template pores and microgels. The size of pore and microgels were measured using the NIS-Elements AR software. The diameter of a template pore was defined as the diameter of the spherical hole instead of the surface pore, as spherical holes determine the microgel size and can easily be measured using emission light mode. For each sample, 9 images from 3 samples were captured randomly, and all pores/particles were measured to collect the diameter data. All the plotting and statistical analysis were completed on Prism 9.

### Microgel preparation efficiency
The pore density of the honeycomb film was defined as the pore count divided by the film area. Nine images of 3 honeycomb films were taken using a Nikon Eclipse Ti2 inverted microscope at ×40. All pores in the frames were manually counted and the frame areas were measured using the software NIS-Elements AR.

The microgels isolated from the templates were collected in 1 mL of Millipore water. To count the number of microgels in the 1 mL suspension, a 5 μL sample was drop-cast on a glass slide and a large image covering the entire droplet was taken using an inverted microscope. The number of microgels in this droplet $n_{5\mu L}$ was then manually counted, and the total amount of microgel was 200 times of $n_{5\mu L}$. For each type of microgel, the procedure was repeated at least four times.

### Fourier transform infrared (FTIR) spectra
FTIR spectra of the phage microgels were represented under by phage hydrogel bulks made with materials exactly same as corresponding microgels. Phage hydrogels were pre-dehydrated, placed in the FTIR Spectrometer (Nicolet 6700, Thermo Fisher Scientific) and the spectra were collected in the range of 4000–500 cm$^{-1}$ using 128 scans at a resolution of 4 cm$^{-1}$.

### Desiccation sensitivity test for phage

A 10 µL drop of phage suspension (M13, HER262 and T7, -10[10] CFU mL⁻¹) was added on a clean, uncovered glass slide at room temperature. The suspension was dried in 10 min and continued to desiccate afterwards. After 1 h, 10 µL of sterile PBS was used to resuspend the phage. The final and original concentrations of phages were titered through full plate plaque assays. The procedure was repeated in triplicate for each type of phage. Phage HER262 was purchased from the Félix d'Hérelle Reference Center for bacterial viruses of the Laval University, and T7 was from Leibniz Institute DSMZ-German Collection of Microorganisms and Cell Cultures.

### Antimicrobial test of phage microgel patches on bacterial lawn

In this experiment, the phage microgels were not isolated from the template. Instead, the composite films were regarded as flexible antimicrobial patches, representing an ordered monolayer of phage microgels. After microgel gelation, the composite patches were washed in sterile water for 20 mins to remove free phages.

Luria-Bertani (LB) agar plates were prepared by suspending LB powder (25 g L⁻¹, Fisher Scientific) in sterile water and supplemented with agar (1.5% w v⁻¹, 15 g L⁻¹, Fischer Scientific) and dispensed into petri dishes (100 × 15 mm, sterile, polystyrene, Fisher Scientific) using a sterile serological pipette. Soft agar overlays were prepared by boiling sterile water supplemented with LB Broth powder (25 g L⁻¹) and agar (0.6% w v⁻¹, 6 g L⁻¹). A 3 mL aliquot of boiled media was dispensed into glass test tubes. Test tubes were then autoclaved to ensure sterility.

Lawns of bacterial overlay were prepared by suspending 100 µL of bacterial suspensions (E. coli ER2738, O157:H7, or BL21) in 3 mL of liquefied soft LB agar, which was vortexed and poured on LB agar plates. After the soft agar was solidified, the washed patches were gently placed on top of the bacterial lawn. The double layer plates were incubated in a stationary incubator (37 °C, VWR International Co.) overnight and subsequently imaged.

### Antimicrobial test for phage microgel sprays on bacterial lawn

One milliliter of the fresh-made phage microgel suspension was transferred into a sprayer. Phage microgel solution was then sprayed on bacterial lawns (prepared as previously described). The double layer agar plates were then incubated in a stationary incubator (37 °C) overnight and subsequently imaged.

### Antimicrobial test for phage microgel suspensions

Two different media, Tryptic Soy Broth (TSB) and nutrient-deficient PBS, were prepared to evaluate the bactericidal ability of phage microgels in different liquid environments.

**Nutrient-rich environment.** Overnight bacterial cultures (E. coli O157:H7 or BL21, -10⁹ CFU mL⁻¹) grown in TSB were diluted to 1:10, 1:100, and 1:1000 in fresh TSB media. For each dilution and the original overnight culture, 10 replicates of 200 µL bacterial solution were added to a sterile 96-well plate. A 10 µL aliquot of the phage microgel suspension was then added to each of the first three replicates as the sample group (labeled "With microgels"). A 10 µL drop of sterile PBS was added to the remaining three replicates as the control group (labeled "No microgels"). Subsequently, the 96-well plate was placed in a microplate reader (Synergy Neo2 Hybrid Multi-Mode Reader, 37 °C, 180 rpm) to measure optical density at a wavelength of 600 nm (OD₆₀₀) every 20 mins for 9 h. Bacterial CFU counts of each replicate were obtained at the end point.

**Nutrient-deficient environment.** Overnight bacterial cultures (E. coli O157:H7 or BL21, -10⁹ CFU mL⁻¹) grown in TSB were diluted to 1:10, 1:100, 1:1000, and 1:10,000 in PBS. For each dilution and the original overnight culture, 10 replicates of 200 µL bacterial solution were added to a sterile 96-well plate. A 10 µL aliquot of the phage microgel

suspension was then added to each of the first three replicates as the sample group (named "With microgels"). A 10 µL drop of sterile PBS was added to the remaining three replicates as the control group (named "No microgels"). Afterwards, the 96-well plate was placed in a shaking incubator (Thermo Scientific, 37 °C, 180 rpm) for 9 h, and the bacterial titer count of each sample at the end point was calculated.

### Food decontamination test of phage microgels

Lettuce (romaine heart) was purchased at the local supermarket and cut into 3 squares weighing 0.4 ± 0.01 g. 2 samples were contaminated with E. coli O157:H7, reaching a contamination level of 10⁶ CFU g⁻¹. A 200 µL aliquot of the phage microgel suspension was then sprayed onto 2 contaminated leaves directly while the other 1 contaminated leaves were sprayed with sterile water. The remaining two leaves served as controls and were wrapped by food wraps without treatment. All 3 lettuce squares were wrapped and placed at room temperature for 9 hrs. The lettuce squares were then immersed in 10 mL of sterile PBS. Then, the samples were unwrapped and immersed in 4 mL of sterile PBS. This mixture was vortexed for 2 min to dislodge bacteria and the titer was determined using standard colony count. MacConkey-Sorbitol ChromoSelect Agar (Millipore Sigma) plates were used for selective O157:H7 titer count[47] (Detection limit: 100 CFU g⁻¹).

The decontamination test for beef steaks (Canadian beef, AAA Angus) followed a similar protocol. Beef steaks were cut into 3 cubes weighing 3 ± 0.1 g. A 30 µL aliquot of E. coli O157:H7 (10⁸ CFU mL⁻¹) was added to 2 meat cubes to achieve a contamination level of 10⁶ CFU g⁻¹. A 200 µL aliquot of the phage microgel suspension was then sprayed onto 1 contaminated meat cubes directly while the other 1 contaminated cubes were sprayed with sterile Millipore water. The remaining 1 cubes served as controls and were wrapped by food wraps without treatment. The 3 meat cubes were placed at room temperature for 9 hrs. The samples were then unwrapped and immersed in 10 mL of sterile PBS. This mixture was vortexed for 2 mins to dislodge the bacteria and bacteria titer was determined using standard colony counts (Detection limit: 34 CFU g⁻¹).

### Reporting summary

Further information on research design is available in the Nature Portfolio Reporting Summary linked to this article.

## Data availability

Data are available from the corresponding author upon request. Source data are provided with this paper.

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

## Acknowledgements

This research was undertaken, in part, thanks to funding from the Canada Research Chairs Program (T.F.D. and Z.H.). L.T., K.J., and Z.H. acknowledge funding from Natural Sciences and Engineering Research Council of Canada (NSERC) Discovery Grants Program and the Boris Family Fund for Health Research. S.K. and K.J. are funded by a Vanier Canada Graduate Scholarship awarded by the Natural Sciences and Engineering Research Council and the Canadian Institute for Health Research respectively.

## Author contributions

L.T. conceived the study, designed and executed the experiments, analyzed the data, prepared the figures, and contributed to writing the manuscript. L.H. and K.J. made important contributions to phage purification, microgel preparation, and antimicrobial tests. A.S. and Z.W. performed some antimicrobial tests and contributed to data analysis. S.K. and T.D. contributed to the food tests. Z.H. conceptualized and supervised the project, and guided the experimental design, data analysis, and manuscript writing.

## Competing interests

All authors declare no competing interests.
