## [Peer Review File · Nature Communications]

Self-Assembling Nanofibrous Bacteriophage Microgels as Sprayable Antimicrobials Targeting Multidrug-Resistant BacteriaREVIEWER COMMENTS

Reviewer #1 (Remarks to the Author):

General description

The manuscript presents a new way to encapsulate the phage in a polymeric matrix, verify its effectiveness and apply the product in a food matrix. The results were satisfactory. However, a few points need to be clarified, but these points does not invalid this work.

Major points

- 1. Abstract.** Is a virus a nanoparticle? I suggest the author to review the concept of nanoparticle, nanomaterial and nanotechnology.
- 2. The abstract needs to be rewritten.** What is the gel matrix? What are the main results?
- 3. Introduction.** Line 18 page three suggests more of a conclusion of the work than an introductory part. I suggest removing or moving that paragraph.
- 4. Methods.** What is the humidity of the film chamber? Was any salt used?
- 5. Many sentences in the result and discussion are similar to the method.** Please remove the result and discussion.

Reviewer #2 (Remarks to the Author):

This is very well written article that reports the fabrication of densely packed phage microgels with a porous fibrous structure. The article is clear and detailed, providing ample information to understand the manipulations performed. The results are well presented and conclusions well substantiated in general by the results. The results achieved are interesting and appear to be novel (although some clarifications are required – see my comments below) and the fibrous structures obtained are well characterized and very appealing. I can foresee several potential applications of these microgels in the biomedical and environmental fields. I recommend publication of this article, after the authors address the few questions and comments below:

- The authors report a glutaraldehyde and an EDC-based methods for crosslinking phages, and that they then mold into honeycomb structures. While the molding strategy may be novel for phage hydrogels, phages have been crosslinked previously with the same crosslinkers. For instance, the Belcher group at MIT has demonstrated the fabrication of GA-crosslinked hydrogels (and aerogels), and EDC-crosslinked thin films (see a few published articles below). It appears like the authors are achieving a high packing density for phages, but can the authors more directly comment on the novelty of their methods and / or the differences in phage density / packing / structure compared with the previously reported crosslinked phage gels from literature?

Highly adjustable 3D nano-architectures and chemistries via assembled 1D biological templates, *Nanoscale* 2019

Assembly of a Bacteriophage-Based Template for the Organization of Materials into Nanoporous Networks, *Adv Mater* 2014

Versatile Three-Dimensional Virus-Based Template for Dye-Sensitized Solar Cells with Improved Electron Transport and Light Harvesting, *ACS Nano* 2013

- Could the authors clarify why the crosslinks between phages do not affect the bioactivity of the phages? What are the important features of the phage for antibacterial activity? Based on antibacterial assays, it appears like BSA is a crucial component of the gel that allows for preserving phage activity – however, the abstract and introduction elude to phage gels without specifically mentioning BSA. If BSA is essential for phage activity, then I think that it should be explicitly mentioned from the beginning of the article that the microgels are composed of phage AND BSA.

- Page 6, lines 20-21: "A single M13 phage exhibits abundant amine and carboxyl groups (5,400 and 10,800, respectively) on its protein coat..." – I assume that the authors are referring to only the reactive subset of amine and carboxylic groups from on the coat proteins of the M13 phage. I believe that many more would be present, but all may not

be reactive. This is not a major comment, but worth clarification by the authors. How were these numbers estimated (from the 2700 copies of the coat protein)?

- Page 7, lines 1-2: "...less than 12 hrs to gel completely..." What is meant by "complete" gelation?

- It is not immediately clear to me what the purpose of the honeycomb molding described on page 7 is. Why was this type of mold select over other molding techniques or mold shapes for the gels?

- In BSA/phage gels, are the same advantages of densely packing phages preserved? Are there tradeoffs to consider between phage-only and BSA-phage hydrogels? Can BSA interfere with some of the potential applications of the phage microgels?

- Based on the FTIR results shown in Figure 3, would the authors be able to calculate a density or ratio of crosslinked amine groups (and carboxylic acid groups) relative to all available groups on the phage?

- The authors rationalize using EDC as alternative crosslinker because of the autofluorescence observed with GA, and the need for gel that do not autofluoresce for "some application scenarios, for example certain biosensing applications that rely on fluorescence to detect target analytes". Could the authors be more specific? How would fluorescent probes eventually be incorporated in the phage hydrogels for these applications?

Reviewer #3 (Remarks to the Author):

The most noteworthy results are the fabrication and potential use of bacteriophage-containing microgels which will add to the existing delivery systems for phages. The methodology is sound, based on established techniques for gelation by crosslinking and characterisation including antimicrobial tests. The manuscript contains sufficient details in the methods for the work to be reproduced. The quality of the work certainly meets the expected standards in the field.

The work is not totally original as the related concept of using filamentous phages to form hydrogels has been explored and reported by the same research team (references 15-17). Hybrid hydrogels of phage M13 and BSA without cross-linkers have already been reported (reference 16), using glutaraldehyde as a widely used protein cross-linker (ref 17). The use of EDC as a crosslinker instead of glutaraldehyde in this paper is thus useful. As the authors stated, application of the established fabrication approach of forming the hexagonal template casting (ref 31,32) is not novel, also was previously reported by the same group, but the novelty lies in the application of the method to prepare phage microgels which is the theme of this paper.

The work does not provide data to support some of the conclusions, eg, on the claims of a high throughput method, as the results focused on the preparation of the microgels and their characterization, instead of study to confirm the high throughput nature of the method. Also, the authors claimed that 'the microgels also protected against desiccation' but there are no long-term stability results to support that claim.

Reviewer 1

The manuscript presents a new way to encapsulate the phage in a polymeric matrix, verify its effectiveness and apply the product in a food matrix. The results were satisfactory. However, a few points need to be clarified, but these points does not invalid this work.

Our response:

We sincerely thank the reviewer for taking the time to assess our work.

Comment #1: Abstract. Is a virus a nanoparticle? I suggest the author to review the concept of nanoparticle, nanomaterial and nanotechnology.

Our response:

Thank you for raising a good point. The concept of terminology is indeed very important. The definition of “nanoparticle” includes fibers and tubes that are less than 100 nm in only two dimensions according to a terminology article recommended by The International Union of Pure and Applied Chemistry (IUPAC).^{1,2} Viruses are acknowledged as “naturally occurring nanoparticles” in multidisciplinary journals such as Nature Nanotechnology and field specific virology journals.^{3–5} Specifically, filamentous phages of *E. coli*, such as M13 used in our research, are nanofilaments with two dimensions at approximately 6.6 nm (**Response Figure 1**). Therefore, M13 phages are commonly referred to as nanoparticles.^{6–8}

Response Figure 1. Schematics of the M13 filamentous phage we use (adapted from *Manuscript Figure 2a*).

Comment #2: The abstract needs to be rewritten. What is the gel matrix? What are the main results?

Our response:

We have rewritten the abstract to highlight the novelties of this work, making the components of the gel and the results more clear:

A. We rewrote the abstract to avoid the misunderstanding that phages are encapsulated in some external gel matrix (*Revised manuscript, page 1*):

*“Here, we **crosslink** half a million self-organized phages as the **sole structural component** to construct each soft microgel”*

B. We rewrote the abstract to better illustrate the results (*Revised manuscript, page 1*):

Problem to solve: *“Nanofilamentous bacteriophages (bacterial viruses) are biofunctional, self-propagating, and monodisperse natural building blocks for virus-built materials. Minifying phage-built materials to microscale offers the promise of expanding the range function for these biomaterials to sprays and colloidal bioassays/biosensors.”*

Product: *“phage-built microgels”*

Approach: *“an in-house developed, biologic-friendly, high-throughput template method”*

Properties: *“self-organized, highly-aligned nanofibrous texture and tunable auto-fluorescence”*

Applications: *“When loaded with potent virulent phages, these microgels effectively reduce heavy loads of multidrug-resistant Escherichia coli O157:H7 on food products, leading to up to 6 logs reduction in 9 hours and rendering food contaminant free.”*

Comment #3: Introduction. Line 18 page three suggests more of a conclusion of the work than an introductory part. I suggest removing or moving that paragraph.

Our response:

We revised this paragraph to better reflect a summary of our work in that context, rather than conclusions of the work (*Revised manuscript, page 3, line 18-23*):

“We demonstrate that two crosslinkers can each effectively assist the gelation of phage nanofilaments through different crosslinking reactions, leading to vastly different fluorescence profiles. We further show that the phage nanofilaments in these virus-exclusive microgels self-assemble into an orderly, highly aligned nanofibrous structure that serve as a high-load delivery vehicle for protein and strong virulent phages to control multidrug-resistant E. coli O157:H7 in food products.”

Comment #4: Methods. What is the humidity of the film chamber? Was any salt used?

Our response:

We forgot to mention this detail in the methods. Thank you for pointing out this oversight. We have added this detail in the method section (*Revised manuscript, page 24, line 20*) to the revised version of the manuscript:

“~55% relative humidity monitored by a humidity sensor”

We originally used a saturated salt solution (Sodium dichromate) to control the humidity in the sealed chamber, but later found that the precipitated salt had corrosive effects on the resinous chamber. Therefore, we refrained controlling humidity with salt and merely added a certain amount of water in the chamber and added a humidity sensor to make sure the humidity remains stable.

Comment #5: Many sentences in the result and discussion are similar to the method. Please remove the result and discussion.

Our response:

We have revised the entire results and discussions section to remove or reword any sentence that bears similarity to sentences in the methods. We have listed the changes here:

- A. Removed the detailed description of breath figure method from the result section (*Revised manuscript, page 7, line 13-16*). It is already illustrated in the Method section (*Revised manuscript, page 24, line 16-23*).
- B. Deleted a few detailed steps of microgel preparation from the result section (*Revised manuscript, page 7, line 20-23*), which are already included in the method section (*Revised manuscript, page 26, line 1-13*).
- C. Deleted the detailed description of how we dehydrated phage microgels from the result section (*Revised manuscript, page 10, line 22-23*), which are already included in the method section (*Revised manuscript, page 26, line 16-18*).

Reviewer 2

This is very well written article that reports the fabrication of densely packed phage microgels with a porous fibrous structure. The article is clear and detailed, providing ample information to understand the manipulations performed. The results are well presented and conclusions well substantiated in general by the results. The results achieved are interesting and appear to be novel (although some clarifications are required – see my comments below) and the fibrous structures obtained are well characterized and very appealing. I can foresee several potential applications of these microgels in the biomedical and environmental fields. I recommend publication of this article, after the authors address the few questions and comments below:

Our response:

Thank you so much for taking the time to assess our work!

Comment #1: The authors report a glutaraldehyde and an EDC-based methods for crosslinking phages, and that they then mold into honeycomb structures. While the molding strategy may be novel for phage hydrogels, phages have been crosslinked previously with the same crosslinkers. For instance, the Belcher group at MIT has demonstrated the fabrication of GA-crosslinked hydrogels (and aerogels), and EDC-crosslinked thin films (see a few published articles below). It appears like the authors are achieving a high packing density for phages, but can the authors more directly comment on the novelty of their methods and / or the differences in phage density / packing / structure compared with the previously reported crosslinked phage gels from literature?

(Highly adjustable 3D nano-architectures and chemistries via assembled 1D biological templates, Nanoscale 2019

Assembly of a Bacteriophage-Based Template for the Organization of Materials into Nanoporous Networks, Adv Mater 2014;

Versatile Three-Dimensional Virus-Based Template for Dye-Sensitized Solar Cells with Improved Electron Transport and Light Harvesting, ACS Nano 2013)

Our response:

This is a great question. We acknowledge (and have cited) previous work on phage-made materials, including the excellent pioneering work from the Belcher lab. Our manuscript reports *five major*

points of novelty, the most outstanding of which is an in-house customized high-throughput mold method to prepare *phage microgels*, which is the theme of this manuscript.

The four other points of novelty are:

- A. The material itself (this is *the first* report on virus-built *microparticles*),
- B. The nanostructure (the high phage packing density resulted in an ordered nanofibrous texture, which had not been reported previously for *phage gels* before, **Response Figure 2**).
- C. Tunable fluorescence dependent on the crosslinker.
- D. The addition of protein into the microgels which prevented detriment of phage bioactivity during crosslinking, and further proposing a diverse microgel spray for food safety.

We now revised the conclusion section to highlight these novelties (*Revised manuscript, page 23, line 11-15*).

“The main discoveries and contributions of our work are (1) establishment of virus-built microparticles, (2) development of a biomolecular-friendly high-throughput preparation method for diverse phage microgels, (3) highly-aligned self-organized nanofibrous texture of phage-exclusive microgels, (4) tunable autofluorescence, and (5) the application of the phage-protein hybrid microgel patch and microgel sprays for biocontrol.”

Below, we have elaborated on the specific points of novelty queried by the reviewer:

A. Method: The developed method here includes 2 aspects: the gelation chemistry and the microgel fabrication process.

A1. Protein bioconjugation chemistry is an important field of research with a rich history, and we do not claim any breakthrough discovery in the gelation chemistry. GA and EDC have been both used to crosslink phages.^{9,10} We would, however, like to point out one new discovery about EDC in our work (*Revised manuscript, page 13-14*), namely that the resulting phage microgels had no obvious fluorescence compared to the strong fluorescence in GA-crosslinked phage microgels. This is useful in any application where background fluorescence is deemed detrimental to performance, e.g., some fluorescence-based biosensors. We did also notice an advantage for EDC as a phage crosslinker, because EDC induces crosslinking by activating the

carboxyl groups on the phage capsid and is thus *not* incorporated into the phage network (Revised manuscript, page 4, line 19-22). This property makes it possible to construct strict “phage-exclusive microgels”. This is not the case for GA.

A2. As for the microgel fabrication process, we customized the preparation method to be high-throughput, heat-free, and solvent-free, all of which are critical for the bioactivity of phage microgels. That is one of the five main novelties in this manuscript as we illustrated above.

B. Phage density, packing, and structure: The high phage packing density we used to construct microgels directly contributed to the highly ordered nanofibrous texture. To the best of our knowledge, this is the first report of such ordered fibrous texture in a phage-made gel, at any scale, whether macro or micro. We have illustrated this in **Response Figure 2**.

Response Figure 2. Comparison of the size and nanostructures of **a.** phage aerogel,¹¹ **b.** phage film,¹² **c.** phage hydrogel⁹ and **d.** phage microgel (current manuscript).

Comment #2: Could the authors clarify why the crosslinks between phages do not affect the bioactivity of the phages? What are the important features of the phage for antibacterial activity? Based on antibacterial assays, it appears like BSA is a crucial component of the gel that allows for preserving phage activity – however, the abstract and introduction elude to phage gels without specifically mentioning BSA. If BSA is essential for phage activity, then I think that it should be explicitly mentioned from the beginning of the article that the microgels are composed of phage AND BSA.

Our response:

This is a great point and we appreciate you bringing it to our attention. Crosslinks between phages do affect the bioactivity of phages, but with the proper addition of proteins, crosslinking will not be detrimental to phage bioactivity. During the crosslinking reaction, the intramolecular crosslinking on the phage capsid, especially on the PIII proteins (bacteria-binding site), could affect the infectious ability of phages (*Supplementary Figure 6*). When added, BSA will react with excessive crosslinker molecules and minimize the intramolecular crosslinking of phages. We have now added the detailed description in the abstract (*Revised manuscript, page 1, line 10-13*) and introduction (*Revised manuscript, page 3, line 20-23*) to make the distinction between “phage-exclusive” and “phage-protein hybrid” microgels clear:

Abstract: “Further preservation of antimicrobial activity was achieved by making hybrid protein-phage microgels.”

Introduction: “We further show that the phage nanofilaments in these virus-exclusive microgels self-assemble into an orderly, highly aligned nanofibrous structure that serve as a high-load vehicle for protein and strong virulent phages to control multidrug-resistant E. coli O157:H7 in food products.”

Comment #3: Page 6, lines 20-21: “A single M13 phage exhibits abundant amine and carboxyl groups (5,400 and 10,800, respectively) on its protein coat...” – I assume that the authors are referring to only the reactive subset of amine and carboxylic groups from on the coat proteins of the M13 phage. I believe that many more would be present, but all may not be reactive. This is not

a major comment, but worth clarification by the authors. How were these numbers estimated (from the 2700 copies of the coat protein)?

Our response:

Correct, the text refers to the reactive subset. We have now added the calculation steps into the supplementary file (*Supplementary Note 1*) to clarify this point. Thank you for pointing out the oversight. The number of reactive subset of amine and carboxylic groups from on the coat proteins were calculated based on the amino acid sequence summarized in reference¹³, as shown in **Response Figure 3**. In the previous manuscript, we missed the alanine at N-terminal in our previous calculation so we incorrectly indicated 5,400. We have now corrected this mistake (*Revised manuscript, page 7, line 1*).

“Supplementary Note 1. The number of reactive functional groups on the M13 phage capsid

The number of reactive subset of amine and carboxylic groups from on the M13 coat proteins were calculated based on the amino acid sequence summarized from reference¹³. M13 capsid is composed of approximately 2700 copies of pVIII protein, and there are reactive amine groups from 2 lysine and 1 alanine (N-terminal) on each pVIII protein. Therefore, there should be approximately 8,100 reactive amine groups (2,700×3).

The reactive carboxyl groups are provided from 2 aspartic acids and 2 glutamic acids in each pVIII protein. Therefore, there are approximately 10,800 reactive carboxyl groups (2,700×4).”

Response Figure 3. Schematic structure of **a.** an M13 bacteriophage and **b.** its major coat proteins. (Figure source from literature¹³)

Comment #4: Page 7, lines 1-2: "...less than 12 hrs to gel completely..." What is meant by "complete" gelation?

Our response:

Thank you for pointing this out. We agree that the word "complete" is not an objective description in this case so we have deleted the word (*Revised manuscript, page 7, line 4*).

Based on our experiences on monitoring the gelation of bulk phage hydrogels, the phage suspension was regarded as having "completely gelled" when it was able to hold the same shape after the mold was inverted (**Response Figure 4**).¹³ In addition, we conformed gelation with a change in rheology.¹³ Based on this method, we found out the M13+EDC bulk hydrogels (same concentration as phage microgels) needs less than 12 hrs to gel.

Response Figure 4. Pictures of GA-cross-linked M13 gels made with different M13 concentrations. (Figure source from literature¹³)

Comment #5: It is not immediately clear to me what the purpose of the honeycomb molding described on page 7 is. Why was this type of mold select over other molding techniques or mold shapes for the gels?

Our response:

Thank you for highlighting this point. We have revised the section on the significance of honeycomb template method (*Revised manuscript, page 7, line 14-16; page 24, line 17-18*).

"This is an easy-approachable and rapid method to fabricate the large-scale template with single-layer, closely-packed, and homogenous micropores without any large equipment."

“This high-throughput method does not require any large equipment or any premade microparticles as templates.”

Response Table 1 summarizes the key significance of our honeycomb molding method compared to other methods used for making microgels (including but not limited to proteinaceous material). You can see from the table that micromolding methods are the only class of methods that are both heat-free and solvent-free, which is critical for preserving phage bioactivity in the microgels. Furthermore, our specific micromolding method, namely polystyrene honeycomb film made via breath figure method, offers three additional advantages:

- A. Easy preparation: The breath figure method does not require any large equipment. We only need to add polystyrene solution into a humid box. After just 20 mins, a polystyrene honeycomb film with ordered and dense pores is automatically prepared.
- B. High-throughput: The honeycomb film we use has 80,000 micropores/cm² and the film size are approximately 5 cm² (the film size can easily be larger as long as we use more polystyrene solution). These films provide 400,000 micropores which is critical for high-throughput preparation of phage microgels. Regular molds made with photoetching or photopolymerization are hard to fabricate micropores to this scale in this short time frame (normally just hundreds to thousands of pores).^{14,15}
- C. Peelable: Microporous polystyrene film can be peeled into 2 layers, which is hard to realize on common molds made with PDMS or rigid materials. The peeling ability is critical to isolate the microgels from the template without using heat or an organic solvent.
- D. Limitation and solution: *“The size and shape of the microparticles is determined to the template pore shape and size. Fortunately, there are already abundant studies extending the breath figure method to fabricate honeycomb films containing ordered pores at different sizes and shapes¹⁶⁻¹⁸.”* (Revised manuscript, page 9, line 9-11)

Response Table 1. Comparison table for different preparation method of microgels

Preparation method	Micromolding		Microfluidics	Emulsion /polymerization	Heat treatment
	Our work	Others			
Building material	Phages (compatible to any environment-sensitive biomaterial)	PEG derivatives ^{14,15}	PEG derivatives ¹⁹ , agarose ²⁰ , alginate ²⁰ , protein ²¹⁻²³	PNIPAM derivatives ²⁴ , PEG derivatives ²⁵	Protein ²⁶⁻²⁸
Compatibility with bioactive molecules	Yes	Yes	Yes	No	No (acid and high heat involved)
Preparation efficiency	High-throughput (Over 35,000 microgels/cm ²)	<10,000 in total	High-throughput	High-throughput	High-throughput
Organic solvent-free	Yes	Yes	No	No	Yes
Gelation time	No requirement	No requirement	Require fast gelation	Require fast gelation	N/A
Microgel size	25 μm (Honeycomb pores can adjust sizes ¹⁷)	Above 100 μm	1 μm to submillimeter	Submicrometer to 100 μm	100 nm to 1 μm
Shape flexibility	Yes (Honeycomb pores can adjust shapes ^{16,18,29})	Yes	No (Spheres and rods)	No (except combining other technics ²⁵)	No
Required large/complex equipment	No	The templates are normally made by lithographic system	Yes	Yes	No

Comment #6: In BSA/phage gels, are the same advantages of densely packing phages preserved? Are there tradeoffs to consider between phage-only and BSA-phage hydrogels? Can BSA interfere with some of the potential applications of the phage microgels?

Our response:

This is a very good point. Advantages and trade-offs of BSA-phage gels are:

Phage+BSA microgels and phage-exclusive microgels we prepared have the same high concentration of phage, but the addition of BSA does result in different properties and potential applications.

The addition of BSA helped preserve the bioactivity of M13 phages but disrupted the phage alignment (*Revised manuscript, Figure 2e*) and decreased the microgel porosity (*Supplementary Figure 6*).

We now added this short discussion topic into the manuscript (*Revised manuscript, page 12, line 3-7*)”

“In conclusion, the phage-exclusive microgels exhibit high porosity, potentiating their strong loading capacity of proteins, phages, and small molecules. The homogenous nanofibrous texture along the same orientation is the direct evidence that the microgels are composed by phages solely crosslinked by small molecule. The addition of protein interfered the order alignment of phages but played an important role in preserving the phage bioactivity which will be illustrated later.”

Potential applications as a result of BSA interference are:

We foresee that phage+BSA microgels and phage-exclusive microgels are suitable for different application scenarios (**Response Table 2**).

Response Table 2. Comparison table for phage-BSA hybrid microgels and phage-exclusive microgels for different applications

	Phage-BSA microgels	Phage-exclusive microgels
Antimicrobial	Yes	No (Crosslinking reaction affects the bioactivity of phages)
Biosensors	Yes (but BSA might block the biosensing agents if they are small molecules)	Yes (Using modified phages as building blocks or adding biosensing molecules)
Drug delivery	Yes	Yes (High porosity can increase drug loading)

Comment #7: Based on the FTIR results shown in Figure 3, would the authors be able to calculate a density or ratio of crosslinked amine groups (and carboxylic acid groups) relative to all available groups on the phage?

Our response:

This is a very interesting idea. To compare crosslinked amine groups with all available groups requires the comparison of the FTIR spectra of the phage solution (or dried phages) with phage microgels (or bulk hydrogels). However, FTIR can be used for quantitative analysis/comparison *only* when the samples are uniform and consistent in volume/size/weight. Therefore, it is hard to make free phage and phage microgels at a comparable level. Considering that the density of crosslinked function groups left is not directly relevant to our claims, we did not add this discussion in the manuscript.

Comment #8: The authors rationalize using EDC as alternative crosslinker because of the autofluorescence observed with GA, and the need for gel that do not autofluoresce for “some application scenarios, for example certain biosensing applications that rely on fluorescence to detect target analytes”. Could the authors be more specific? How would fluorescent probes eventually be incorporated in the phage hydrogels for these applications?

Our response:

This is a great discussion topic that we are actively working on. We aim to incorporate fluorescent aptamers in our gels for sensing of analytes³⁰. We are working on developing aptamers that exhibit suitable chemistry for incorporation into our gels which can help realize bacteria-sensing and bacteria-killing at the same time. Therefore, low fluorescence background as a result of EDC cross linking can significantly improve the sensitivity of these sensors.

Reviewer 3

“The most noteworthy results are the fabrication and potential use of bacteriophage-containing microgels which will add to the existing delivery systems for phages. The methodology is sound, based on established techniques for gelation by crosslinking and characterisation including antimicrobial tests. The manuscript contains sufficient details in the methods for the work to be reproduced. The quality of the work certainly meets the expected standards in the field.”

Our response:

We sincerely thank the reviewer for taking the time to assess our work.

Comment #1: The work is not totally original as the related concept of using filamentous phages to form hydrogels has been explored and reported by the same research team (references 15-17). Hybrid hydrogels of phage M13 and BSA without cross-linkers have already been reported (reference 16), using glutaraldehyde as a widely used protein cross-linker (ref 17). The use of EDC as a crosslinker instead of glutaraldehyde in this paper is thus useful. As the authors stated, application of the established fabrication approach of forming the hexagonal template casting (ref 31,32) is not novel, also was previously reported by the same group, but the novelty lies in the application of the method to prepare phage microgels which is the theme of this paper.

Our response:

This is a shrewd observation. The reviewer has correctly pinpointed the level of novelty in each aspect of our work. We agree that the microgel preparation method and usage of EDC (which realized fluorescence control) are indeed important contributions to the field. In addition, we would like to point out the following regarding the novelty of our work:

1. The material itself: the phage microgels we built are the first *virus-built microparticles* (no other material with the same composition and at this scale have been reported); comparing to bulk hydrogels, microgels have a broader range of applications as sprayable antimicrobials, in drug-delivery vehicles, and biosensing/bioassays.
2. The nanostructure: The high packing density for phages we used to construct microgels directly contributed to the ordered nanofibrous texture. To the best of our knowledge, such ordered fibrous texture has not been shown on any *phage gels* before, and most certainly not at the scale of our microgels (**Response Figure 3**).

Comment #2: The work does not provide data to support some of the conclusions, eg, on the claims of a high throughput method, as the results focused on the preparation of the microgels and their characterization, instead of study to confirm the high throughput nature of the method. Also, the authors claimed that ‘the microgels also protected against desiccation’ but there are no long-term stability results to support that claim.

Our response:

This is a very good point, thank you.

A. Claim of high-throughput:

High-throughput is normally defined as testing/screening/preparing/testing of 10,000 to 100,000 compounds per day, either through manual operation or automated systems^{31,32}. We now added the description in the manuscript where we reported the amount of obtained microgels (*Revised manuscript, page 10, line 11-13*).

“In summary, we obtained over 3.5×10^4 phage microgels from every square centimeter of our template. Every film we made was over 5 cm^2 , allowing for the production of 175,000 phage microgels in a single day. In addition, more than 10 films can be applied to produce microgels simultaneously, demonstrating the high-throughput ability of this method. Each microgel contains more than 3.8×10^5 phage particles, constituting a phage community of 10^{10} in total.”

Based on these numbers, we claim that our method is high-throughput.

B. Claim of desiccation control:

We have now added the data supporting this small claim (*Supplementary Figure 7*). As shown in **Response Figure 5**, free HER262 phages (1×10^6 PFU/mL) lost bioactivity significantly after air-drying for 1 hour while same amount of HER262 within microgels remain significantly more infective through the same process.

Response Figure 5. Bioactivity of 1×10^6 PFU/mL of free HER262 phages and M13+HER262+BSA+GA hybrid phage microgels (before and after air-drying for 1 hr) on *E. coli* O157:H7 lawn.

Format Changes of Figure Design

Figure 1:

We enlarged *Manuscript Figure 1a-b* and added light blue background to present better the crosslinking reactions and the schematic of microgel preparation, as shown in **Response Figure 6**. Same blue background was also used in *Manuscript Figure 2b* now.

Response Figure 6. Revisions we made in *Revised Manuscript Figure 1a-b* to enlarge the schematics of crosslinking reactions and microgel preparation method.

Figure 4:

We now provided new schematics of phage HER262 and T7 in *Revised Manuscript Figure 4a*, as shown in **Response Figure 7**. Corresponding schematics were updated in *Supplementary Figure 9-10* as well.

Response Figure 7. *Manuscript Figure 4a* before and after revision.

References:

1. IUPAC Recommendations Published From 2010 to 2015. *The International Union of Pure and Applied Chemistry (IUPAC)* (2015). Available at: <https://iupac.org/recommendation/iupac-recommendations-published-from-2010-to-2015/>. (Accessed: 18th July 2022)
2. Vert, M. *et al.* Terminology for biorelated polymers and applications. *Pure Appl. Chem.* **84**, 377–410 (2012).
3. Nanotechnology versus coronavirus. *Nat. Nanotechnol.* **15**, 617 (2020).
4. Park, J. S. *et al.* A highly sensitive and selective diagnostic assay based on virus nanoparticles. *Nat. Nanotechnol.* **4**, 259–264 (2009).
5. Nkanga, C. I. & Steinmetz, N. F. The pharmacology of plant virus nanoparticles. *Virology* **556**, 39–61 (2021).
6. Passaretti, P., Khan, I., Dafforn, T. R. & Goldberg Oppenheimer, P. Improvements in the production of purified M13 bacteriophage bio-nanoparticle. *Sci. Rep.* **10**, 1–9 (2020).
7. Ksendzovsky, A. *et al.* Convection-enhanced delivery of M13 bacteriophage to the brain: Laboratory investigation. *J. Neurosurg.* **117**, 197–203 (2012).
8. Steinmetz, N. F. Viral nanoparticles as platforms for next-generation therapeutics and imaging devices. *Nanomedicine Nanotechnology, Biol. Med.* **6**, 634–641 (2010).
9. Peivandi, A., Tian, L., Mahabir, R. & Hosseinidou, Z. Hierarchically Structured, Self-Healing, Fluorescent, Bioactive Hydrogels with Self-Organizing Bundles of Phage Nanofilaments. *Chem. Mater.* **31**, 5442–5449 (2019).
10. Courchesne, N. M. D. *et al.* Assembly of a bacteriophage-based template for the organization of materials into nanoporous networks. *Adv. Mater.* **26**, 3398–3404 (2014).
11. Jung, S. M., Qi, J., Oh, D., Belcher, A. & Kong, J. M13 Virus Aerogels as a Scaffold for Functional Inorganic Materials. *Adv. Funct. Mater.* **27**, 1603203 (2017).
12. Chen, P. Y. *et al.* Versatile three-dimensional virus-based template for dye-sensitized solar cells with improved electron transport and light harvesting. *ACS Nano* **7**, 6563–6574 (2013).
13. Chung, W. J., Lee, D. Y. & Yoo, S. Y. Chemical modulation of M13 bacteriophage and its functional opportunities for nanomedicine. *Int. J. Nanomedicine* **9**, 5825–5836 (2014).
14. Li, Y. *et al.* Rapid Assembly of Heterogeneous 3D Cell Microenvironments in a Microgel Array. *Adv. Mater.* **28**, 3543–3548 (2016).
15. Yeh, J. *et al.* Micromolding of shape-controlled, harvestable cell-laden hydrogels. *Biomaterials* **27**, 5391–5398 (2006).
16. Daly, R., Sader, J. E. & Boland, J. J. Taming Self-Organization Dynamics to Dramatically Control Porous Architectures. *ACS Nano* **10**, 3087–3092 (2016).
17. Takehiro Nishikawa *et al.* Fabrication of Honeycomb Film of an Amphiphilic Copolymer at the Air–Water Interface. *Langmuir* **18**, 5734–5740 (2002).
18. Wang, W. *et al.* Deterministic Reshaping of Breath Figure Arrays by Directional Photomanipulation. *ACS Appl. Mater. Interfaces* **9**, 4223–4230 (2017).
19. Headen, D. M., Aubry, G., Lu, H. & García, A. J. Microfluidic-based generation of size-controlled, biofunctionalized synthetic polymer microgels for cell encapsulation. *Adv. Mater.* **26**, 3003–3008 (2014).
20. Eydelnant, I. A., Betty Li, B. & Wheeler, A. R. Microgels on-demand. *Nat. Commun.* **5**, 1–9 (2014).
21. Roode, L. W. Y., Shimanovich, U., Wu, S., Perrett, S. & Knowles, T. P. J. Protein

- Microgels from Amyloid Fibril Networks. *ACS Nano* **11**, 223–263 (2019).
22. Shimanovich, U., Song, Y., Brujic, J., Shum, H. C. & Knowles, T. P. J. Multiphase protein microgels. *Macromol. Biosci.* **15**, 501–508 (2015).
 23. Zhou, X. M. *et al.* Enzymatically Active Microgels from Self-Assembling Protein Nanofibrils for Microflow Chemistry. *ACS Nano* **9**, 5772–5781 (2015).
 24. Brugger, B. & Richtering, W. Magnetic, thermosensitive microgels as stimuli-responsive emulsifiers allowing for remote control of separability and stability of oil in water-emulsions. *Adv. Mater.* **19**, 2973–2978 (2007).
 25. An, H. Z., Helgeson, M. E. & Doyle, P. S. Nanoemulsion composite microgels for orthogonal encapsulation and release. *Adv. Mater.* **24**, 3838–3844 (2012).
 26. Schmitt, C. *et al.* Internal structure and colloidal behaviour of covalent whey protein microgels obtained by heat treatment. *Soft Matter* **6**, 4876–4884 (2010).
 27. Nicolai, T. Formation and functionality of self-assembled whey protein microgels. *Colloids Surfaces B Biointerfaces* **137**, 32–38 (2016).
 28. Phan-Xuan, T. *et al.* On the crucial importance of the pH for the formation and self-stabilization of protein microgels and strands. *Langmuir* **27**, 15092–15101 (2011).
 29. Zhu, C., Tian, L., Liao, J., Zhang, X. & Gu, Z. Fabrication of Bioinspired Hierarchical Functional Structures by Using Honeycomb Films as Templates. *Adv. Funct. Mater.* **28**, 1–8 (2018).
 30. Jepsen, M. D. E. *et al.* Development of a genetically encodable FRET system using fluorescent RNA aptamers. *Nat. Commun.* **9**, 1–10 (2018).
 31. Inglese, J. *et al.* High-throughput screening assays for the identification of chemical probes. *Nat. Chem. Biol.* **3**, 466–479 (2007).
 32. Shang, L., Cheng, Y. & Zhao, Y. Emerging Droplet Microfluidics. *Chem. Rev.* **117**, 7964–8040 (2017).

REVIEWERS' COMMENTS

Reviewer #1 (Remarks to the Author):

The manuscript presents a new way to encapsulate the phage in a polymeric matrix, verify its effectiveness and apply the product in a food matrix. The results were satisfactory and the review answered all my questions.

Reviewer #3 (Remarks to the Author):

Comments sufficiently addressed.